# Wiskott Aldrich syndrome protein regulates non-selective autophagy and mitochondrial homeostasis in human myeloid cells

Elizabeth Rivers[1,2], Rajeev Rai[1], Jonas Lötscher[3], Michael Hollinshead[4], Gasper Markelj[5], James Thaventhiran[6], Austen Worth[2], Alessia Cavazza[1], Christoph Hess[3], Mona Bajaj-Elliott[1], Adrian J Thrasher[1,2]*

[1]Infection, Immunity and Inflammation Programme, University College London Great Ormond Street Institute of Child Health, London, United Kingdom; [2]Department of Immunology, Great Ormond Street Hospital for Children NHS Foundation Trust, London, United Kingdom; [3]Department of Biomedicine, Immunobiology, University of Basel, Basel, Switzerland; [4]Department of Pathology, University of Cambridge, Cambridge, United Kingdom; [5]Department of Allergy, Rheumatology and Clinical Immunology, University Children's Hospital, University Medical Centre Ljubljana, Ljubljana, Slovenia; [6]Medical Research Council-Toxicology Unit, School of Biological Sciences, University of Cambridge, Cambridge, United Kingdom

**\*For correspondence:**
a.thrasher@ucl.ac.uk

**Competing interests:** The authors declare that no competing interests exist.

**Abstract** The actin cytoskeletal regulator Wiskott Aldrich syndrome protein (WASp) has been implicated in maintenance of the autophagy-inflammasome axis in innate murine immune cells. Here, we show that WASp deficiency is associated with impaired rapamycin-induced autophagosome formation and trafficking to lysosomes in primary human monocyte-derived macrophages (MDMs). WASp reconstitution in vitro and in WAS patients following clinical gene therapy restores autophagic flux and is dependent on the actin-related protein complex ARP2/3. Induction of mitochondrial damage with CCCP, as a model of selective autophagy, also reveals a novel ARP2/3-dependent role for WASp in formation of sequestrating actin cages and maintenance of mitochondrial network integrity. Furthermore, mitochondrial respiration is suppressed in WAS patient MDMs and unable to achieve normal maximal activity when stressed, indicating profound intrinsic metabolic dysfunction. Taken together, we provide evidence of new and important roles of human WASp in autophagic processes and immunometabolic regulation, which may mechanistically contribute to the complex WAS immunophenotype.

## Introduction

The Wiskott Aldrich syndrome protein (WASp) is an essential regulator of the actin cytoskeleton in haematopoietic cells. It co-ordinates cytoskeletal dynamics important in immune cell functions, through recruitment of the actin-related protein complex ARP2/3 and formation of new branched actin filaments (reviewed in *Rivers and Thrasher, 2017*).

Wiskott Aldrich syndrome (WAS), caused by loss of function of WASp, is a rare X-linked disorder resulting in significant combined immune deficiency, microthrombocytopaenia and susceptibility to haematological malignancy. Symptoms of classical WAS present in infancy and most children will die from disease-related complications in the first 15 years of life, in the absence of definitive treatment

by haematopoietic stem cell transplantation (HSCT) or gene therapy. An inflammatory phenotype occurs in up to 80% of children, with symptoms such as severe eczema, inflammatory bowel disease (IBD), arthritis, and vasculitis, which may persist after HSCT in a proportion of children, particularly where myeloid chimerism is low (*Worth and Thrasher, 2015*).

We recently demonstrated disturbance of the autophagy-inflammasome axis in WASp-deficient murine innate immune cells (*Lee et al., 2017*). Inflammasomes are cytosolic protein complexes responsible for maturing the pro-inflammatory cytokines IL-1β and IL-18 and promoting inflammatory cell death, pyroptosis (*Martinon et al., 2002*). Assembling in response to endogenous and/or exogenous danger signals, they co-ordinate and aid in mounting appropriate immunity against potential threats, such as from invading pathogens (*Guo et al., 2015*; *Franchi et al., 2009*). Autophagy is a lysosomal degradation pathway crucial for maintaining cellular homeostasis and downregulating inflammasomes, through clearance of inflammasome triggers (*Shi et al., 2012*; *Harris et al., 2011*; *Rodgers et al., 2014*). Autophagosomes are double-membraned vesicles that form around cytoplasmic contents and are transported to lysosomes, where fusion allows for content degradation (*Shibutani et al., 2015*). Autolysosomal contents are subsequently recycled either to the cytoplasm, or transported to the cell surface for antigen presentation and lysosomes are re-formed ready for further autophagosome interactions (*Yu et al., 2010*).

Some autophagic processes are non-selective in nature, for example the catabolic degradation of bulk cytoplasmic contents to provide metabolic substrates following nutrient starvation ('macroautophagy'). Others involve selective degradation, for example of damaged organelles, for example mitochondria ('mitophagy') or pathogens that have escaped phagocytosis ('xenophagy'). These require a highly co-ordinated process, whereby targets for degradation must first be identified and recruit autophagy receptors.

Actin is involved in autophagy at several levels, including autophagosome formation, maturation, and transport to lysosomes [reviewed in *Kast and Dominguez, 2017*; *Kruppa et al., 2016*; *Coutts and La Thangue, 2016*]. The ARP2/3 complex, through which WASp acts, consists of seven subunits (ARP2, ARP3 and ARPC1-5) and has been particularly implicated (*Kast et al., 2015*; *Coutts and La Thangue, 2015*; *Xia et al., 2013*; *Xia et al., 2014*; *King et al., 2013*; *Zavodszky et al., 2014*; *Zhang et al., 2016*). In an ex vivo model of *Enteropathogenic Escherichia coli* (EPEC) infection in murine bone-marrow-derived dendritic cells (BMDCs), we provided the first demonstration of a crucial role for WASp in xenophagy, where WASp deficiency resulted in the absence of canonical autophagosome formation (*Lee et al., 2017*). In this model, impaired autophagy contributed to exaggerated inflammasome activity presumably due to defective clearance of intracellular bacteria.

To date, the potential role of WASp in regulating autophagy in human immune cells remains unstudied. Here, we provide evidence of an essential role for WASp in both non-selective autophagy and selective mitophagy in the human myeloid compartment, where it appears to be a novel player in the maintenance of mitochondrial homeostasis. We propose that metabolic consequences found in WASp deficiency identify WAS as a disorder of immunometabolic regulation, thus highlighting new potential therapeutic pathways for further exploration.

## Results

### WASp plays a pivotal role in autophagosome formation following non-selective autophagy stimulation

We first investigated the role of human WASp in non-selective autophagy in the human monocytic cell line, THP-1. *WAS* KO THP-1 cells were generated by CRISPR gene-editing (*Figure 1—figure supplement 1a–c*). Rapamycin inhibits the mTOR complex initiating autophagy, which culminates in LC3I to LC3II conversion on autophagosome membranes, the latter process being a well-established surrogate marker of autophagosome formation.

WT and *WAS* KO THP-1 cells were exposed to rapamycin and LC3II protein expression monitored through western blotting. In WT THP-1 cells, an increase in LC3II expression was observed (*Figure 1a*). In contrast, LC3II expression did not increase in the absence of WASp, leading to a significant reduction in LC3II expression in *WAS* KO compared with WT cells (*Figure 1a and b*). Given that autophagy stimulation results in a flux of autophagosomes, we assessed whether this difference

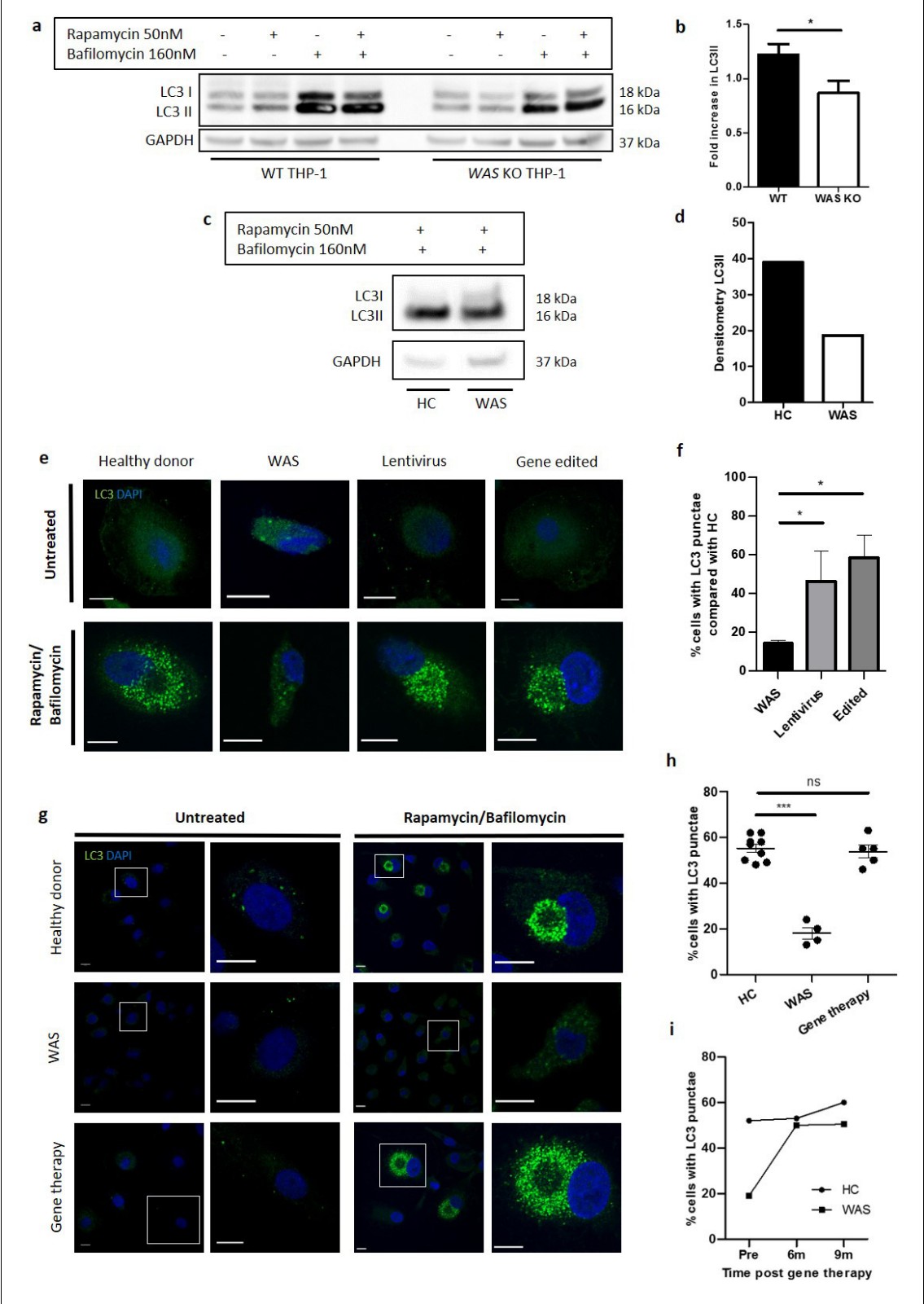

**Figure 1.** WASp is necessary for autophagosome formation. (a) WT and *WAS* KO THP-1 were cultured with rapamycin 50 nM +/- bafilomycin 160 nM for 6 hr and cell lysates immunoblotted for LC3I and LC3II with GAPDH as loading control. Representative blot from five independent experiments. (b) Combined densitometric analysis from (a). Histogram of fold increase in LC3II from baseline control normalised to GAPDH loading control following rapamycin treatment. Bars represent mean +/- SEM (n = 5 in duplicate). Two-tailed t-test *p<0.05 (c). MDMs from a healthy donor and WAS patient

*Figure 1 continued on next page*

*Figure 1 continued*

cultured with rapamycin 50 nM and bafilomycin 160 nM for 6 hr and cell lysates immunoblotted for LC3I and LC3II with GAPDH as loading control. (d) Densitometric analysis of (c). Histogram of LC3II densitometry normalised to GAPDH loading control following rapamycin and bafilomycin treatment (n = 1 in duplicate). (e) Representative images of SDMs from a healthy donor, WAS patient and in vitro corrected WAS patient SDMs using lentiviral transduction or gene editing techniques cultured with rapamycin 50 nM and bafilomycin 160 nM for 6 hr. Cells fixed and stained for LC3 (green) and nuclei (DAPI, blue). Imaged by confocal microscopy at 63x. Scale bar represents 10 μm. Details of the level of WASp expression and gene marking in the experiments summarised in e and f can be found in *Figure 1—source data 2*. (f) SDMs from healthy donor, WAS patient and in vitro corrected WAS patient SDMs using gene therapy or gene editing techniques cultured with rapamycin 50 nM and bafilomycin 160 nM for 6 hr. Cells fixed and stained for LC3. Slides blinded for imaging and analysis. Cells forming significant LC3 punctae were counted from confocal microscopy images taken at 20x. 300–1000 cells per slide analysed from six fields of view. Histogram displays percentage of cells with LC3 punctae compared with their matched healthy donors from the same experiment with bars representing median +/- IQR. Combined analysis from four independent experiments. Mann Whitney *p<0.05. (g) Representative images of MDMs from a healthy donor (top panel), WAS patient (middle panel), and WAS patient after treatment with gene therapy (lower panel) cultured with rapamycin 50 nM and bafilomycin 160 nM for 6 hr. Cells fixed and stained for LC3 (green) and nuclei (DAPI, blue). Imaged by confocal microscopy at 63x (first column), with further detail of areas highlighted by white boxes shown to the right. Scale bars represent 10 μm. (h) MDMs from healthy donors (n = 5), WAS patients (n = 3) and WAS patients after treatment with gene therapy (n = 2) cultured with rapamycin 50 nM and bafilomycin 160 nM for 6 hr. Cells fixed and stained for LC3 (green) and nuclei (DAPI, blue). Slides blinded for imaging and analysis. Cells forming significant LC3 punctae were counted from confocal microscopy images taken at 20x. 300–1000 cells per slide analysed from six fields of view. Dot plot of percentage of cells forming LC3 punctae displayed, with points representing independent experiments. Bars represent mean +/- SEM. Two-tailed t-test ***p<0.0001. (i) MDMs from healthy donors and the same WAS patient before and after gene therapy treatment cultured with rapamycin 50 nM and bafilomycin 160 nM for 6 hr. Cells fixed and stained for LC3 (green) and nuclei (DAPI, blue). Slides blinded for imaging and analysis. Cells forming significant LC3 punctae were counted from confocal microscopy images taken at 20x. 300–1000 cells per slide analysed from six fields of view. Percentage of cells forming LC3 punctae displayed. DAPI, 4′,6-diamidino-2-phenylindole; GAPDH, glyceraldehyde 3-phosphate dehydrogenase; HC, healthy control; IQR, interquartile range; MDM, monocyte-derived macrophage; ns, not significant; SDM, stem-cell-derived macrophage; SEM, standard error of the mean; WAS, Wiskott Aldrich syndrome.

The online version of this article includes the following source data and figure supplement(s) for figure 1:

**Source data 1.** Molecular details of patient monocyte-derived macrophages used in experiments.
**Source data 2.** Molecular details of patient stem-cell-derived macrophages used for in vitro WAS correction WASp, Wiskott Aldrich syndrome protein.
**Figure supplement 1.** WASp is necessary for autophagosome formation.

in LC3II abundance reflected abrogated autophagosome formation, or an increase in autophagosome degradation. Bafilomycin inhibits autophagosome-lysosome fusion, preventing lysosomal degradation of autophagosomes. Co-culture with the combination of rapamycin and bafilomycin confirmed that reduced LC3II expression observed in *WAS* KO THP-1 cells was indicative of impaired autophagosome formation rather than increased degradation (*Figure 1a*). This observation was confirmed in primary human monocyte-derived macrophages (MDMs) derived from a WAS patient compared with a healthy donor (*Figure 1c and d*).

Confocal microscopic analysis of healthy donor MDMs demonstrated that the increase in LC3II expression in response to rapamycin/bafilomycin treatment correlated with an increase in LC3 punctae in the perinuclear region (*Figure 1—figure supplement 1d and e*). Further analysis by electron microscopy (EM) revealed that these LC3 punctae represent autophagosomes, which gather around the centrosomes (*Figure 1—figure supplement 1f*).

To explore the role of WASp in LC3 punctae formation, primary haematopoietic stem and progenitor cell (HSPC)-derived macrophages (SDMs) and MDMs from healthy donors and WAS patients were exposed to rapamycin/bafilomycin and subjected to analysis by confocal microscopy. WAS SDMs and MDMs showed significant reduction in LC3 punctae formation when compared with healthy donors (*Figure 1e–h*), supporting the hypothesis that autophagosome formation is indeed impaired in the absence of WASp.

To add further credence to our initial findings, we investigated the impact of in vitro and in vivo WAS correction on autophagosome formation. Primary HSPCs from WAS patients underwent in vitro WASp reconstitution by delivery of a lentiviral vector expressing *WAS* (lentiviral transduction) or by CRISPR/Cas9-mediated targeted integration of a correct copy of *WAS* (gene editing) and were differentiated to macrophages. Rapamycin-induced autophagy of these *WAS*-corrected SDMs demonstrated significantly greater LC3 punctae formation compared with uncorrected SDMs from the same WAS patients (*Figure 1e and f*). These findings were confirmed following in vivo WASp reconstitution in WAS patients treated with clinical gene therapy, where rapamycin stimulation of MDMs from WAS patients treated with gene therapy demonstrated LC3 punctae formation comparable to

healthy controls (*Figure 1g and h*). Additionally, we were able to follow one patient through gene therapy, repeating rapamycin stimulation of MDMs from the same patient before and after treatment, with restoration of LC3 punctae observed at 6 and 9 months after engraftment of corrected HSPCs (*Figure 1i*).

Collectively, our observations suggest for the first time that WASp plays an important role in autophagosome formation in response to non-selective autophagy in human myeloid cells.

## Human WASp is implicated in delivery of autophagosomes to lysosomes in non-selective autophagy

In addition to reduced number, the cellular location of the LC3 punctae in WAS SDMs and MDMs differed when compared with heathy donors. Instead of peninuclear clustering, LC3 punctae in WAS SDMs and MDMs were scattered throughout the cytoplasm (*Figure 1e and g*).

This observation led us to postulate that autophagosomes clustering around the perinuclear region may aid fusion with lysosomes. To test our hypothesis, LC3 and LAMP-1 (a lysosomal membrane marker) co-localisation in primary MDMs was investigated. In resting healthy donor MDMs, few LC3 punctae were seen scattered throughout the cytoplasm (*Figure 2a*). In contrast, LAMP-1 staining showed that lysosomes are more plentiful and found predominantly in the perinuclear region at rest. Upon rapamycin-induced autophagy, LC3 punctae markedly increased in number and co-localised with LAMP-1 punctae in the perinuclear region. EM analysis revealed that this co-localisation in the perinuclear region was around the microtubule organising centres (*Figure 1—figure supplement 1f*). Taken together, the data suggest that autophagosomes are formed in the cytoplasm and travel along microtubules for lysosomal fusion. In contrast to healthy donors, WAS patient MDMs revealed disorganised co-localisation between LC3 and LAMP-1, with more than one cluster of LC3 and LAMP-1 punctae in addition to uncoordinated punctae scattered throughout the cytoplasm (*Figure 2a*). The disorganised appearance of LC3 and LAMP-1 punctae in the absence of WASp, supports our hypothesis for the importance of WASp in autophagosome-lysosome traffic and fusion.

In models of in vitro WAS correction, we found restoration of LC3 punctae organisation following rapamycin-induced autophagy of SDMs (*Figure 1e*), further supporting a WASp-dependent role in autophagosome dynamics. Co-staining of LC3 and LAMP-1 in MDMs from two WAS patients after gene therapy also revealed restoration of LC3 and LAMP-1 co-ordination, comparable to healthy donors (*Figure 2a*). It is important to note that in addition to disrupted LC3 and LAMP-1 punctae co-ordination, a significant reduction in LAMP-1 punctae was noted in WAS patient MDMs compared with healthy donors, both at rest and following autophagy stimulation. This was confirmed by reduced protein expression of LAMP-1 in WASp-deficient THP-1 cells and primary MDMs (*Figure 2b–e*). Through EM analysis, we observed an abundance of single-membraned tubular structures in healthy donor MDMs, which were surprisingly reduced in WAS patient MDMs (*Figure 2—figure supplement 1*). Although tubular lysosomes have previously been reported in macrophages (*Swanson et al., 1987*), without further staining it is difficult to say with confidence whether these tubular structures represent lysosomes or not. Collectively, these observations could indicate a potential role for WASp in lysosome formation, or a reduction in basal autophagic flux in WAS, since an important source of lysosomes is from recycling of material from autophagolysosomes.

## The role of WASp in non-selective autophagy is ARP2/3-dependent

We sought to identify a potential mechanism(s) that would explain the role of WASp in our chosen model of non-selective autophagy. Confocal analysis of healthy donor MDMs and SDMs exposed to rapamycin demonstrated close association of actin with LC3 (*Figure 3a* and *Figure 3—figure supplement 1a*). More detailed analysis using deconvolution of super resolution confocal images revealed that in some instances there was also true co-localisation, with actin rings directly overlapping rings of LC3 (*Figure 3—figure supplement 1a and b*). Actin localisation with LC3 was impaired in WAS patient MDMs (*Figure 3a and b*) and SDMs (*Figure 3—figure supplement 1c*), but restored following WASp reconstitution either in vitro (*Figure 3—figure supplement 1c*) or in vivo (*Figure 3a and b*), suggesting that the role of WASp in these cellular processes is largely actin-dependent.

We also explored the role of ARP2/3 in this model. Healthy donor MDMs treated with CK666 (a chemical inhibitor of ARP2/3 complex activity), prior to rapamycin treatment, lost their ability to

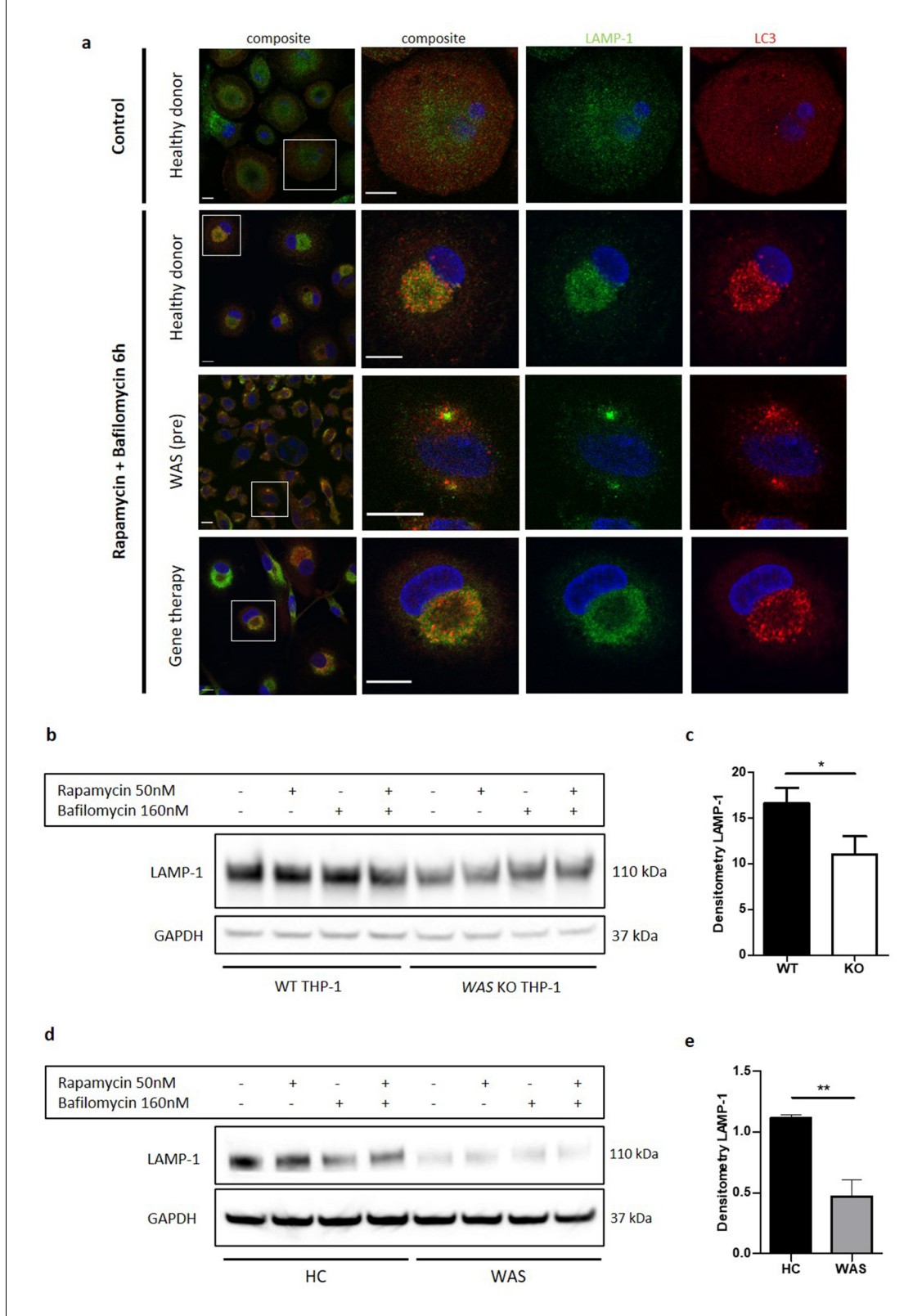

**Figure 2.** WASp is necessary for delivery of autiophagosomes to lysosomes.  (a) MDMs from a healthy donor, WAS patient and WAS patient following treatment with gene therapy cultured with or without rapamycin 50 nM and bafilomycin 160 nM as indicated for 6 hr. Cells fixed and stained for LC3 (red), LAMP-1 (green), and nuclei (DAPI, blue). Imaged by confocal microscopy at 63x, with further magnification of selected area highlighted by white box. Scale bar 10 μm. Representative images from five healthy donors, two WAS patients and two WAS patients post gene therapy. (b) WT and *WAS*

eLife Research article

Cell Biology | Immunology and Inflammation

KO THP-1 cultured with rapamycin 50 nM +/- bafilomycin 160 nM for 6 hr as indicated. Cell lysates immunoblotted for LAMP-1 expression, with GAPDH as loading control. Representative blot from three independent experiments. (c) Combined densitometric analysis from (b). Histogram shows densitometry of LAMP-1 normalised to GAPDH loading control from unstimulated WT and *WAS* KO THP-1 cells (n = 3 in duplicate). Bars represent mean +/- SEM. Two-tailed t-test *p<0.05. (d) CD14[+] PBMCs from healthy donors or WAS patients cultured with rapamycin 50 nM +/- bafilomycin 160 nM for 1.5 hr as indicated. Cell lysates immunoblotted for LAMP-1 expression, with GAPDH as loading control. Representative blot from three independent experiments in duplicate. (e) Combined densitometric analysis from (d). Histogram shows densitometry of LAMP-1 normalised to GAPDH loading control for unstimulated healthy donor and WAS patient MDMs. Bars represent mean +/- SEM (n = 3 in duplicate). Two-tailed t-test **p<0.01 DAPI, 4',6-diamidino-2-phenylindole; GAPDH, glyceraldehyde 3-phosphate dehydrogenase; HC, healthy control; MDM, monocyte-derived macrophage; PBMCs, peripheral blood mononuclear cells; SEM, standard error of the mean; WAS, Wiskott Aldrich syndrome.
The online version of this article includes the following figure supplement(s) for figure 2:

**Figure supplement 1.** WASp is necessary for autophagosome formation.

form LC3 punctae, strikingly recapitulating WASp-deficiency (*Figure 3c and d*). ARPC1B is one of the seven subunits comprising the ARP2/3 complex. Patients with defects in this subunit exhibit a clinical phenotype largely similar to WAS (*Kahr et al., 2017*; *Kopitar et al., 2019*), suggesting the two conditions share underlying mechanism(s). At present, whether these proteins play a role in autophagy is as yet unexplored. Since little is known about the cellular defects of ARP2/3 subunit deficiencies, we first sought to confirm an ARP2/3 functional defect in patient cells. Podosomes are established WASp-ARP2/3-dependent actin-rich structures important in adhesion and migration of myeloid cells, absent in WAS and CK666-treated healthy MDMs (*Figure 3—figure supplement 1e*). Primary MDMs from ARPC1B-deficient patients were also unable to form podosomes (*Figure 3—figure supplement 1d*). Furthermore, induction of non-selective autophagy in ARPC1B-deficient MDMs resulted in impaired LC3 punctae formation (*Figure 3c and d*), LC3-actin association (*Figure 3b*), and LC3II expression (*Figure 3e and f*), as seen in WAS. Taken together, our observations suggest that during non-selective autophagy in human MDMs and SDMs, actin associates with LC3 punctae in a WASp and ARP2/3-dependent manner.

To explore further whether the role of WASp in autophagy is dependent on ARP2/3, we used the iLIR web resource platform developed by Kalvari et al, to investigate the likelihood of an LC3-interacting region (LIR) in WASp (*Kalvari et al., 2014*).

Analysis of WASp's amino acid sequence did not find any xLIR domains (*Figure 3—source data 1*). Several WxxL motifs were identified, but similarity scores (PSSM scores) were low (max. 10), making a functional interaction between WASp and LC3 unlikely. The same process was carried out for all seven ARP2/3 complex subunits. In contrast to WASp analysis, three out of the seven ARP2/3 subunits (ARP2, ARP3, and ARPC2) were found to contain xLIR motifs, with high PSSM scores, highly predictive of functional LIR domains. Additionally, ARPC1B was identified to have a WxxL motif with high PSSM score that could also be predicted to contain a functional LIR motif.

## WASp-dependent actin cages encapsulate damaged mitochondria in mitophagy

Selective autophagy is a highly co-ordinated process, whereby targets for degradation are first identified and recruit autophagy receptors prior to being sequestered by autophagosomes. Our previous work demonstrated a marked defect in xenophagy in murine models of WASp deficiency, herein we sought to explore effects on similar pathways in human cells. Models of xenophagy are complicated as several cellular events including bacterial clearance through phagocytosis, autophagy and a hybrid process known as LC3-associated phagocytosis (LAP), all contribute to early bacterial sensing and processing. With this in mind, we turned our attention to the role of WASp in mitophagy, a non-infection model of selective autophagy.

Mitochondrial morphology is complex, with mitochondria often forming branched networks rather than existing as discrete organelles (*Hollinshead et al., 1997*). The protonophore carbonyl cyanide 3-chlorophenylhydrazone (CCCP) is routinely employed to investigate mitophagy (*Kruppa et al., 2018*; *Villa et al., 2017*; *Hsieh and Yang, 2019*). By depolarising the mitochondrial membrane, CCCP treatment results in fragmentation of the mitochondrial network (without causing cell death; *Figure 4—figure supplement 1b*), with damaged mitochondria being targeted for degradation by mitophagy. Anti-biotin antibody has been validated for specific mitochondrial imaging,

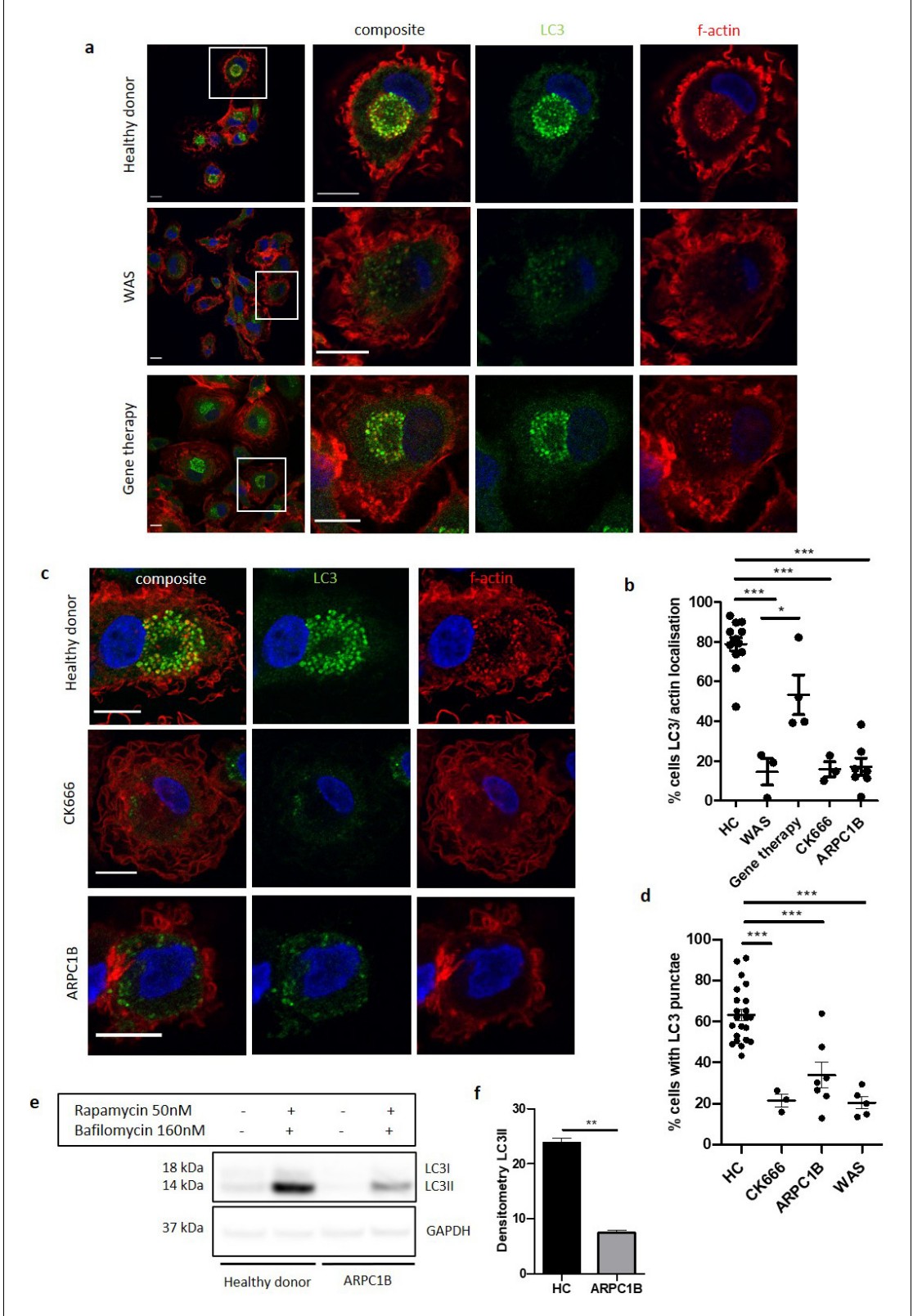

**Figure 3.** The role of WASp in non-selective autophagy is ARP2/3-dependent. (a) Representative images of MDMs from a healthy donor, WAS patient and WAS patient post gene therapy cultured with rapamycin 50 nM and bafilomycin 160 nM for 6 hr. Cells fixed and stained for LC3 (green), f-actin (phalloidin, red) and nuclei (DAPI, blue). Imaged by confocal microscopy at 63x, with further magnification of highlighted area inside white boxes. Scale bar = 10 μm (b) MDMs from healthy donors (n = 7), WAS patients (n = 2), WAS patients post gene therapy (n = 2), healthy donor MDMs treated with

*Figure 3 continued on next page*

Figure 3 continued

CK666 20 µM (n = 2) and ARPC1B (n = 4)-deficient patients cultured with rapamycin 50 nM and bafilomycin 160 nM for 6 hr. Cells fixed and stained for LC3 (green), f-actin (phalloidin, red) and nuclei (DAPI, blue). Imaged by confocal microscopy at 63x and percentage of cells with LC3 and actin localisation calculated from 50 to 100 cells per slide from at least three fields of view. Dots represent independent experiments and bars represent mean +/- SEM. Two-tailed t-test *p<0.05, ***p<0.0001. (c) Representative images of MDMs from healthy donors, CK666 20µM-treated healthy donors and ARPC1B-deficient patients cultured with rapamycin 50 nM and bafilomycin 160 nM for 6 hr. Cells fixed and stained for LC3 (green), f-actin (phalloidin, red), and nuclei (DAPI, blue). Imaged by confocal microscopy at 63x. Scale bar = 10 µm. (d) MDMs from healthy donors (n = 7), CK666 20µM-treated healthy donors (n = 2) and ARPC1B (n = 4)-deficient patients cultured with rapamycin 50 nM and bafilomycin 160 nM for 6 hr. Cells fixed and stained for LC3. Imaged by confocal microscopy at 20x and cells forming significant LC3 punctae analysed from 500 to 1000 cells from at least six fields of view per slide. Dots represent independent experiments with bars representing mean +/- SEM. Two-tailed t-test ***p<0.0001. (e) MDMs from two healthy controls and two ARPC1B-deficient patients cultured with rapamycin 50 nM and bafilomycin 160 nM for 6 hr. Cell lysates immunoblotted for LC3I and LC3II expression, with GAPDH for loading control. Representative blot from two independent experiments in duplicate. (f) Combined densitometric analysis from (e). Histogram shows densitometry of LC3II expression normalised to GAPDH loading control from rapamycin and bafilomycin treated MDMs (n = 2 independent experiments in duplicate). DAPI, 4′,6-diamidino-2-phenylindole; GAPDH, glyceraldehyde 3-phosphate dehydrogenase; HC, healthy control; MDM, monocyte-derived macrophage; SEM, standard error of the mean; WAS, Wiskott Aldrich syndrome.

The online version of this article includes the following source data and figure supplement(s) for figure 3:

**Source data 1.** Summary of LC3-interating regions of human WASp and all seven subunits of the ARP2/3 complex as identified using the web resource developed by Kalvari et al.

**Figure supplement 1.** The role of WASp in non-selective autophagy is ARP2/3 dependent.

taking advantage of the high presence of biotinylated enzymes within mitochondria (**Hollinshead et al., 1997**), a protocol utilised in the current study. Following CCCP treatment, healthy donor MDMs demonstrated marked disruption of the mitochondrial network (**Figure 4a**). When bafilomycin was used in combination with CCCP treatment (to prevent mitophagosome degradation), clusters of mitochondria were found to be surrounded by actin cages, suggesting successful identification of damaged mitochondria and isolation from the healthy network. In some instances, the network was completely disrupted, with only fragments of mitochondria visible within actin cages (**Figure 4—figure supplement 1a**). Localisation of LC3 to these actin cages confirms that encapsulated mitochondria were targeted for degradation by mitophagy.

In marked contrast to healthy donors, WAS MDMs demonstrated a lack of mitochondrial clustering, with significantly impaired actin cage formation (**Figure 4a and b**). Where present, actin cages were small and demonstrated reduced LC3 recruitment (**Figure 4—figure supplement 1a and c**). In vivo WAS correction, through clinical gene therapy, restored mitophagy (**Figure 4a and b**), indicating that the formation of actin cages around damaged mitochondria is a WASp-dependent process. Furthermore, inhibiting ARP2/3 activity of healthy donor MDMs led to significant disruption of actin cage formation after induction of mitochondrial damage. Similarly, mitophagy induction in primary ARPC1B-deficient MDMs revealed impaired actin cage formation, thereby mimicking the WASp-deficient phenotype (**Figure 4b and c**).

In summary, the present study provides the first evidence of an important role for human WASp in mitophagy, which is dependent on activation of the ARP2/3 complex.

## WASp is essential for maintaining mitochondrial homeostasis

Mitophagy is integral to mitochondrial homeostasis and increasing evidence suggests that actin is crucial for mitochondrial fission and fusion (**Li et al., 2015**; **Hatch et al., 2014**; **Moore et al., 2016**). With this in mind, we next sought to identify the impact of WASp-deficiency on maintenance of the basal mitochondrial network. At rest, healthy donor MDMs demonstrated two main categories of mitochondrial morphology:networked and fragmented (**Figure 5a**). In the majority, mitochondria existed in networks, with a small percentage of fragmented morphology (**Figure 5b and c**). Co-staining for f-actin revealed the presence of these networks at the basal cell surface, with actin foci, consistent with podosomes, observed in close proximity.

Unlike healthy donors, the majority of mitochondria in WAS MDMs were fragmented and where networks were present, these were found throughout the cell cytoplasm rather than at the basal cell surface (**Figure 5b and c**). After gene therapy, the basal mitochondrial morphology of corrected WAS MDMs returned to a prominent basal location, akin to healthy controls (**Figure 5b and c**). This suggests that, even in the absence of additional mitochondrial insult, basal mitochondrial

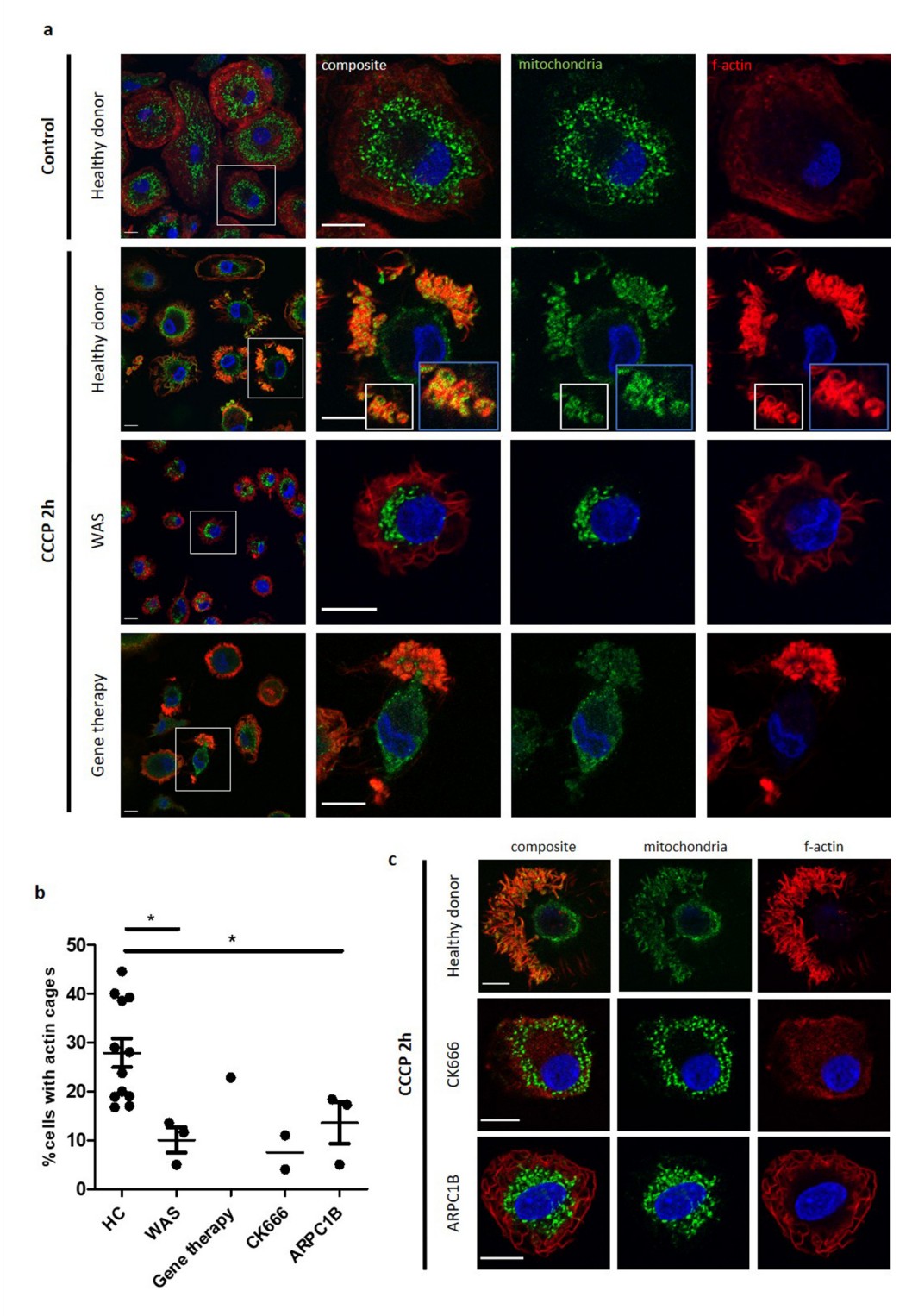

**Figure 4.** WASp plays an ARP2/3-dependent role in mitophagy. (**a**) MDMs from a healthy donor, WAS patient and WAS patient post gene therapy cultured with or without CCCP 10 μM (plus bafilomycin 160 nM) for 2 hr. Cells fixed and stained for mitochondria (biotin, green), f-actin (phalloidin, red) and nuclei (DAPI, blue) and imaged by confocal microscopy. Representative images illustrated at 63x, with higher magnification of areas highlighted by white boxes. Actin cages further magnified from healthy donor, denoted by blue boxes (inset). Scale bars = 10 μm. (**b**) Healthy donor (n = 7), WAS patient (n = 2), WAS patient post-gene therapy (n = 1), CK666 100μM-treated healthy donor (n = 1) and ARPC1B-deficient patient (n = 3) MDMs cultured with CCCP 10 μM (plus bafilomycin 160

*Figure 4 continued on next page*

*Figure 4 continued*

nM) for 2 hr. Cells fixed and stained for mitochondria and f-actin. Slides blinded prior to imaging and analysis. At least six fields of view per slide were imaged at 63x by confocal microscopy. Cells forming actin cages around mitochondria were analysed from at least 100 cells per slide. Dots represent independent experiments, with bars denoting mean +/- SEM. Two-tailed t-test *p<0.05. (c) Representative images of healthy donor, CK666 100μM-treated healthy donor and ARPC1B-deficient MDMs cultured with CCCP 10 μM (plus bafilomycin 160 nM) for 2 hr. Cells fixed and stained for mitochondria (biotin, green), f-actin (phalloidin, red), and nuclei (DAPI, blue) and imaged by confocal microscopy at 63x. Scale bars = 10 μm. CCCP, carbonyl cyanide m-chlorophenylhydrazone; DAPI, 4',6-diamidino-2-phenylindole; HC, healthy control; MDM, monocyte-derived macrophage; SEM, standard error of the mean; WAS, Wiskott Aldrich syndrome.

The online version of this article includes the following figure supplement(s) for figure 4:

**Figure supplement 1.** WASp plays an ARP2/3-dependent role in mitophagy.

homeostasis is compromised. This is a significant finding as it suggests for the first time that WASp may play a vital role in maintenance of a healthy mitochondrial network.

ARPC1B-deficient and ARP2/3-inhibited healthy donor MDMs also demonstrated perturbed mitochondrial morphology (*Figure 5c and d*). Inhibition of ARP2/3 in healthy donor MDMs with CK666 led to fragmentation of mitochondrial networks, sharing the WAS phenotype. Basal mitochondrial morphology of ARPC1B-deficient MDMs was also disrupted, but subtle differences were noted. As with WAS and CK666-treated MDMs, a reduction of networked mitochondria was seen in ARPC1B-deficient MDMs, compared with healthy donor MDMs; however, this did not reach statistical significance. Interestingly, however, the morphology of networked mitochondria in ARPC1B-deficient MDMs appeared more elongated compared with healthy donor MDMs (*Figure 5d*). The functional significance of this observation is at present unknown, but it is tempting to speculate that ARPC1B may play a specific role in mitochondrial fission, through WASp-independent pathways. Collectively, our observations provide novel evidence that WASp plays a crucial role in maintenance of mitochondrial homeostasis, which is also ARP2/3-dependent.

## WASp deficiency is associated with impaired mitochondrial function

Finally, we sought to identify potential functional consequences of the observed abnormalities of mitophagy and mitochondrial morphology in WAS. To this end, oxygen consumption rate (OCR), a measure of mitochondrial oxidative phosphorylation (OxPhos), was measured in healthy donor and WAS MDMs, using metabolic flux analyses. In these experiments, WAS MDMs exhibited significant reduction in basal mitochondrial respiration and trend towards reduced maximal respiration (*Figure 6a and b*). Concomitant analysis of extracellular acidification rates (ECAR), reflecting aerobic glycolysis, found no significant difference between healthy donor and WAS MDMs (*Figure 6c and d*), suggesting no compensatory switch to aerobic glycolysis as described for other cell types in the absence of autophagy (*Clarke and Simon, 2019*). The inability of WAS MDMs to respond to impaired OxPhos by upregulating aerobic glycolysis raises the possibility that metabolic undersupply could lead to downstream functional cellular consequences.

## Discussion

Although the importance of the actin cytoskeleton in autophagy is being increasingly recognised, this phenomenon remains less well studied in primary human cells. Building on our previous findings identifying a key role for murine WASp in the autophagy-inflammasome axis, we now provide evidence that human WASp is important in both non-selective autophagy and selective mitophagy. Gaining mechanistic insights into the pathophysiology of WAS allows us to report for the first time some important metabolic consequences of WASp deficiency, which may eventually help to clarify some previously unexplained features of the complex disease immunophenotype. Furthermore, with increasing interest in the importance of cellular metabolism in immune outcome in health and disease, our findings may have far-reaching implications for other diseases of cytoskeletal dysfunction.

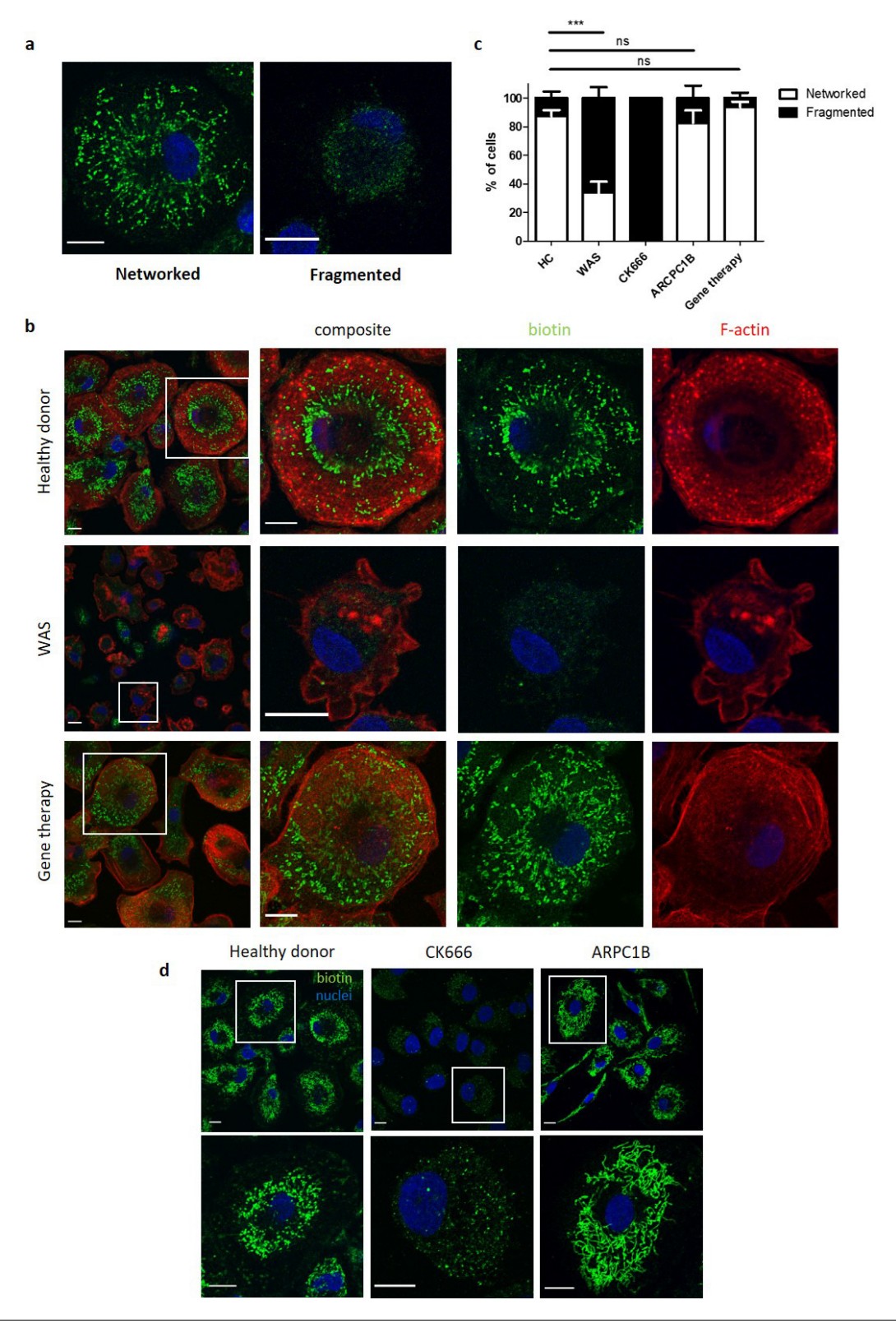

**Figure 5.** WASp is necessary for mitochondrial homeostasis. (a) Representative images of different mitochondrial morphologies from healthy donor MDMs fixed and stained for mitochondria (biotin, green) and nuclei (DAPI, blue). Imaged by confocal microscopy at 63x. Scale bar = 10 µm. (b) Representative images of healthy donor (n = 7), WAS patient (n = 3) or WAS patient post gene therapy (n = 2) MDMs fixed and stained for mitochondria (biotin, green), f-actin (phalloidin, red) and nuclei (DAPI, blue). Imaged at 63x by confocal microscopy, with higher magnification of area

*Figure 5 continued on next page*

*Figure 5 continued*

inside white boxes displayed to the right. Scale bar = 10 µm. (c) Combined analysis from (b and d). Slides blinded prior to imaging and analysis. At least 100 cells per slide analysed for mitochondrial morphology and categorised according to appearance as indicated. Bars represent mean +/- SEM. (d) Representative images of healthy donor MDMs (n = 7), CK666 100µM-treated healthy donor MDMs (n = 2), or MDMs from ARPC1B-deficient patients (n = 3) fixed and stained for mitochondria (biotin, green) and nuclei (DAPI, blue). Imaged at 63x by confocal microscopy and higher magnification of area highlighted by white boxes shown to the right. Scale bar = 10 µm. DAPI, 4',6-diamidino-2-phenylindole; HC, healthy control; MDM, monocyte-derived macrophage; SEM, standard error of mean; WAS, Wiskott Aldrich syndrome.

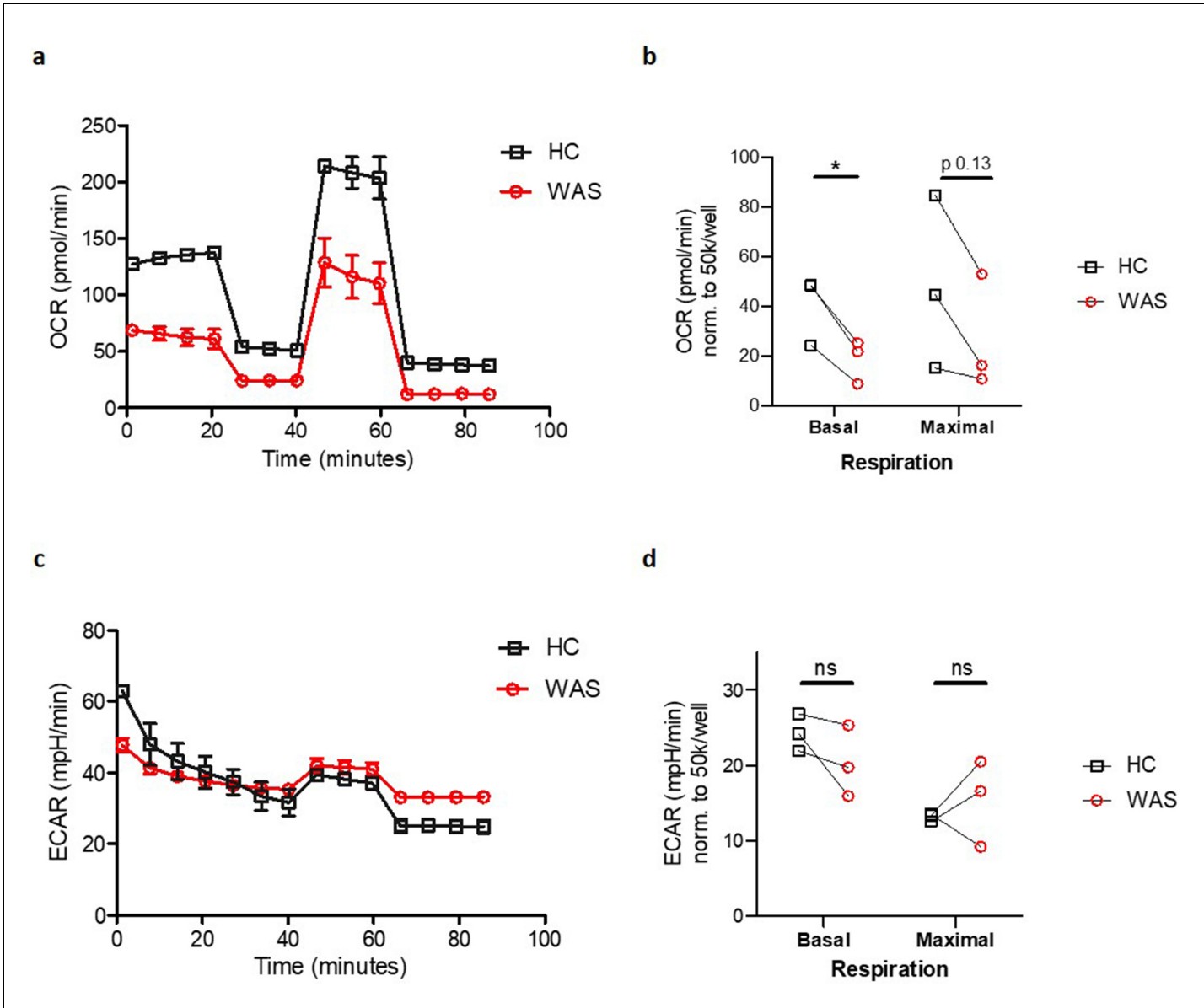

**Figure 6.** WASp deficiency is associated with impaired mitochondrial function. (a) Oxygen consumption rate (OCR) of MDMs from healthy donors and WAS patients during Mito Stress test. Representative plot from three independent experiments in triplicate. (b) Summary of OCR from three independent experiments, normalised to $50 \times 10^3$ cells/well, with OCR at basal and maximal respiration from WAS patient MDMs paired to their respective healthy controls (HC). Paired t-test *p<0.05, ns = not significant. (c) Extracellular acidification rates (ECAR) of MDMs from healthy donors and WAS patients during Mito Stress test. Representative plot from three independent experiments in triplicate. (d) Summary of ECAR from three independent experiments, normalised to $50 \times 10^3$ cells/well, with ECAR at basal and maximal respiration from WAS patient MDMs paired to their respective healthy controls (HC). Paired t-test ns = not significant.

In the present study, the potential contribution of WASp to both non-selective autophagy and selective mitophagy in human myeloid cells was investigated. A significant role for human WASp in autophagosome formation and dynamics was uncovered. Although WASp-deficiency resulted in only modest reduction in LC3II expression in THP-1 and primary MDMs, this is in keeping with our previous findings in murine BMDCs following rapamycin-induced autophagy (*Lee et al., 2017*). Importantly, in xenophagy, our previous work demonstrated that this same modest reduction in LC3II expression correlated with complete absence of canonical autophagosome formation around intracellular bacteria (*Lee et al., 2017*). This finding highlights that expression of LC3II is not exclusive to mature autophagosomes, but may also associate with immature autophagosomes and other single-membraned vesicles. Here, we show that reduced LC3II expression in THP-1 cells and primary WAS MDMs correlates with a significant reduction in LC3 punctae, a surrogate marker for autophagosome formation, which are restored following in vitro and in vivo WAS correction.

Our recent work was the first to identify a role for murine WASp in autophagy (*Lee et al., 2017*), although the mechanism(s) of action remained unexplored. In the present study, we elucidated that the mechanism of human WASp action in non-selective autophagy and mitophagy is dependent on ARP2/3-mediated actin polymerisation. Actin is seen to colocalise with LC3 punctae in rapamycin-induced autophagy of healthy donor SDMs and MDMs, which is abrogated in WAS, ARP2/3-inhibited healthy donors and ARPC1B-deficient patients. Restoration of actin-LC3 punctae association following in vitro and in vivo WAS correction confirms a WASp-dependent role in this model. Identification of potential LIR motifs in ARP2/3 components, not present in WASp, adds further support to the notion that the role of WASp in autophagy is via ARP2/3 actin polymerisation; however, it is worth also noting that LIR-independent modes of interaction with Atg-8 family proteins have also been reported (*Behrends et al., 2010*). Impaired autophagy in ARP2/3-inhibited and ARPC1B-deficient patients, also highlight a potential WASp-independent role for ARP2/3 in autophagy. Interestingly, reduction in LC3II protein expression following rapamycin-induced autophagy of ARPC1B-deficient patients, compared with healthy donors, was more profound than that seen in WAS. This may reflect WASp-independent polymerisation of actin via the ARP2/3 complex, which may be of particular importance in potential for compensatory mechanisms to restore some autophagic activity essential for cell survival in WASp deficiency. Our findings here are supported by extensive work by others in non-haematopoietic cell line models that highlight the importance of actin at various stages of the autophagy process (reviewed in *Kast and Dominguez, 2017*; *Coutts and La Thangue, 2016* and *Kruppa et al., 2016*), for which ARP2/3 and WASp family members have been particularly implicated (*Kast et al., 2015*; *Coutts and La Thangue, 2015*; *Xia et al., 2013*; *Xia et al., 2014*; *King et al., 2013*; *Zavodszky et al., 2014*; *Zhang et al., 2016*). To our knowledge, however, this is the first study reporting the importance of WASp and ARP2/3 in autophagy of primary human haematopoietic cells.

In addition to the role of actin in autophagosome formation and dynamics, existing evidence demonstrates that the transport of mature autophagosomes to lysosomes in non-haematopoietic cell lines relies on microtubules (*Jahreiss et al., 2008*; *Kimura et al., 2008*). Our EM and confocal analyses of primary MDMs following rapamycin-induced autophagy identified clustering of autophagosomes around microtubule-organising centres. Although previous evidence to support a role of WASp in microtubule homeostasis is limited predominantly to a study where WASp was overexpressed in Cos7 cells (*Tian et al., 2000*), our data suggests in autophagy there may be a direct role of WASp in microtubule-dependent trafficking of autophagosomes. The mechanism of this link and to what degree actin may also contribute is at present unclear and requires further elucidation.

We were surprised to note that there appeared to be a reduction in lysosome numbers in untreated WAS MDMs. This could indicate an additional role for WASp in lysosome formation, or reflect impaired basal autophagic flux, since lysosomes re-form in late autophagy following content degradation of the autophagolysosomes (*Yu et al., 2010*). Functionally, this observation may be important in antigen presentation, a role previously described for lysosomes in macrophages (*Vyas et al., 2007*; *Saric et al., 2016*).

Mitochondria are essential in providing energy for cellular homeostasis and play an important role in innate immunity, contributing to pathogen control through bactericidal activity of mitochondrial reactive oxygen species (mtROS) (*West et al., 2011*). However, ROS can also lead to damage of cellular organelles, including the mitochondria themselves. Mitophagy plays a crucial role in ensuring damaged mitochondria are effectively cleared in order to maintain mitochondrial network

integrity and cellular homeostasis. In a model of CCCP-induced mitophagy of primary MDMs, we found sites of actin polymerisation surrounding mitochondria that had become detached from the mitochondrial network, encapsulating them in cage-like structures. Actin-cage formation around damaged mitochondria was abrogated in WAS, ARP2/3-inhibited and ARPC1B-deficient patient MDMs and restored in WAS patients after clinical gene therapy. This suggests that, as in non-selective autophagy, mitophagy is dependent on the ARP2/3 activity of WASp. Our work here is supported by a recent study where ARP2/3-dependent actin polymerisation was identified to be important in mitophagic actin cage formation in HEK293 cells (*Kruppa et al., 2018*). Additionally, mitophagic actin cage formation demonstrated in our study here bore similarity to the WASp-dependent actin-septin cages seen in our previous study investigating EPEC-mediated xenophagy (*Lee et al., 2017*). To our knowledge, we are now the first to report an ARP2/3-dependent role for WASp in mitophagy of primary human MDMs.

Furthermore, in the absence of mitochondrial insult we have surprisingly uncovered a significant disruption to basal mitochondrial morphology in WAS MDMs, suggesting a role for WASp in maintenance of mitochondrial homeostasis. Since actin is important in mitochondrial homeostasis through fission and fusion (*Li et al., 2015*; *Hatch et al., 2014*; *Moore et al., 2016*), it is not clear whether disrupted mitochondrial morphology in the absence of WASp reflects defective mitophagy alone or other as yet unexplored cellular events. Interestingly, the appearance of fragmented mitochondria, as seen in WAS MDMs, has been shown by others to be associated with caspase-1-mediated inflammasome activation in BMDCs (*Park et al., 2015*). This could provide a further link between our findings here of disrupted mitochondrial morphology and previous evidence of inflammasome dysregulation in WAS innate immune cells even in the absence of pathogen challenge (*Lee et al., 2017*).

Exploring the functional consequences of disrupted mitochondrial networks in WAS MDMs uncovered profound defects in mitochondrial respiration. Impaired basal mitochondrial respiration in WAS is likely to have a significant impact on cellular functions, through lack of ATP production. Additionally, we also discovered reduced maximal respiratory capacity, meaning that when WASp-deficient MDMs become 'stressed', they are unable to increase their respiratory drive in order to meet the required energy demands. Macrophage mitochondria are critical players in the metabolic response to pathogens and cell stress, as they can switch from OxPhos metabolism to glycolysis (*O'Neill and Pearce, 2016*; *Escoll et al., 2017*). Interestingly, our ECAR analysis revealed that the observed differences in OxPhos are not explained by an upregulation of glycolysis in WAS, with similar rates of glycolysis in healthy donor and WASp-deficient MDMs.

The discovery of a metabolic defect in WAS MDMs is, to our knowledge, the first description of its kind. However, the idea of a metabolic defect in WAS platelets was inferred as early as 40–50 years ago, with several studies observing a paucity of mitochondria and impaired mitochondrial energy production (*Trung et al., 1975*; *Verhoeven et al., 1989*; *Akkerman et al., 1982*; *Obydennyi et al., 2020*). Although as yet unstudied, extending these metabolic defects to other immune cell lineages may go some way to explaining other immunophenotypic features of WAS including T cell anergy, progressive reduction in B and T lymphocytes with age and increasing development of autoinflammatory symptoms. This is supported by work in non-WAS models, following emerging evidence of the role of immune cell metabolism in autophagy and differentiation (*Clarke and Simon, 2019*; *Bantug et al., 2018b*). Immune cells particularly susceptible to metabolic dysfunction in the presence of impaired autophagy include T regulatory cells, as well as mature T and B cells, which rely more heavily on OxPhos for their differentiation (*Clarke and Simon, 2019*; *Wei et al., 2016*; *Price et al., 2018*; *Clarke et al., 2018*; *Chen et al., 2014*; *Bantug et al., 2018a*).

Our exploration of the role of human WASp in autophagy began by means of evaluating the mechanism of inflammation in WAS, with a view to identifying novel candidate therapeutic targets. Autophagy is important in down-regulating inflammasome activity through effective clearance of intracellular debris such as pathogens or damaged organelles. Here, we have shown for the first time that human WASp is a key player in both non-selective autophagy and selective mitophagy. Moreover, we have uncovered a significant defect in mitochondrial homeostasis with important metabolic consequences and which may have implications well beyond the enhanced inflammatory phenotype of WAS.

# Materials and methods

## Key resources table

| Reagent type (species) or resource | Designation | Source or reference | Identifiers | Additional information |
|---|---|---|---|---|
| Cell line (*Homo sapiens*) | WT THP-1 | ATCC | TIB-202 | Male, 1 year infant |
| Cell line (*Homo sapiens*) | WAS KO THP-1 | doi: 10.1038/ s41467-017- 01676-0 | | Male, 1 year infant |
| Antibody | Anti-human WASP (Mouse monoclonal) | BD Bioscience | Cat. #: 557773, RRID:AB_396867 | FACS (1:100), WB (1:500) |
| Antibody | Anti-mouse IgG Alexa Fluor 647 (Goat monoclonal) | BioLegend | Cat. #: 405301, RRID:AB_315005 | FACS (1:200) |
| Antibody | Anti-human LAMP-1 (Mouse monoclonal) | BD Bioscience | Cat. #: 17–611043; RRID:AB_3983356 | WB (1 in 1000) |
| Antibody | Anti-human LC3B (Rabbit monoclonal) | Sigma | Cat. #: L7543; RRID:AB_796155 | WB (one in1000) |
| Antibody | Anti-human GAPDH (Mouse monoclonal) | Santa Cruz | Cat. #: sc365062; RRID:AB_10847862 | WB (1 in 1000) |
| Antibody | Anti-mouse IgG HRP (Sheep monoclonal) | GE Healthcare | Cat. #: NA9310-1ML; RRID:AB_772193 | WB (1 in 2000) |
| Antibody | Anti-rabbit IgG HRP (Donkey monoclonal) | GE Healthcare | Cat. #: NA9340-1ML; RRID:AB_772191 | WB (1 in 2000) |
| Antibody | Anti-human LC3 (Rabbit polyclonal) | MBL | Cat. #: PM036; RRID:AB_2274121 | IF (1 in 200) |
| Antibody | Anti-human LAMP-1 (Mouse monoclonal) | CST | Cat. #: 15665; RRID:AB_2798750 | IF (1 in 50) |
| Antibody | Anti-human Vinculin (Mouse monoclonal) | Sigma | Cat. #: V4505; RRID:AB_477617 | IF (1:200) |
| Antibody | Anti-mouse IgG Alexa Fluor 488 (Goat polyclonal) | Molecular Probes | Cat. #: A32723; RRID:AB_2633275 | IF (1 in 500) |
| Antibody | Anti-rabbit IgG Alexa Fluor 546 (Goat polyclonal) | Molecular Probes | Cat. #: A-11035; RRID:AB_143051 | IF (1 in 500) |
| Antibody | Anti-rabbit IgG Alexa Fluor 647 (Goat polyclonal) | Molecular Probes | Cat. #: A27040; RRID:AB_2536101 | IF (1 in 500) |
| peptide, recombinant protein | Fibronectin | R and D system | Cat. #: 4305-FNB-200 | |
| commercial assay or kit | CD14+ Microbead | Miltenyi Biotec | Cat. #: 130-050-201 | |
| Chemical compound, drug | Rapamycin | Calbiochem | Cat. #: 553211 | |

*Continued on next page*

*Continued*

| Reagent type (species) or resource | Designation | Source or reference | Identifiers | Additional information |
|---|---|---|---|---|
| Chemical compound, drug | Bafilomycin A1 | Sigma | Cat. #: SML1661 | |
| Chemical compound, drug | CK666 | Abcam | Cat. #: ab141231 | |
| Chemical compound, drug | CCCP | Merck | Cat. #: 215911 | |
| Chemical compound, drug | Oligomycin | Sigma | Cat. #: 75351 | |
| Chemical compound, drug | FCCP | Sigma | Cat. #: C2920 | |
| Chemical compound, drug | Rotenone | Sigma | Cat. #: R8875 | |
| Software, algorithm | ImageJ | NIH | RRID:SCR_003070 | |
| Software, algorithm | Prism | GraphPad | Version 8 | |
| Software, algorithm | iLIR | doi: 10.4161/auto.28260 | Version 1 | |
| Other | Prolong Diamond anti-fade mounting solution with DAPI | Molecular probes | Cat. #: P36962 | |
| Other | Phalloidin-633 | Molecular Probes | Cat. #: A22284 | |

## Cells and cell lines

WAS KO THP-1 were generated as previously described (*Lee et al., 2017*). KO was confirmed with Sanger sequencing, WASp expression, and functional WASp analysis of podosome formation (*Figure 1—figure supplement 1a–c*). Cells were maintained in RPMI 1640 with Glutamax (Gibco) containing 10% FBS (Sigma) and 1% penicillin/streptomycin (Gibco), at 37°C with 5% $CO_2$ and regularly screened for mycoplasma contamination. Cells were differentiated to macrophages using 10 ng/ml PMA (Sigma) for 24 hr, followed by rest for 24 hr in medium prior to experiments.

For MDMs, peripheral blood samples were obtained from healthy adult volunteers and patients with WAS and ARPC1B deficiencies (*Figure 1—source data 1*), following informed written consent (REC 06/Q0508/16). Peripheral blood mononuclear cells (PBMCs) were isolated from whole blood using Ficoll-Histopaque gradient separation and $CD14^+$ cells positively selected using magnetic microbead separation (Miltenyi 130-042-201), according to manufacturer's instructions. Monocyte-derived macrophages were obtained by culturing in RPMI (supplemented as above), containing M-CSF (Gibco) 20 ng/ml for 5–7 days.

## Gene editing of WAS HSPCs

To restore WASp expression by CRISPR/Cas9-mediated site-specific integration of a *WAS* cDNA, patient-derived WAS HSPCs (*Figure 1—source data 2*) were manipulated as described in *Rai et al., 2020*. Briefly, $0.2 \times 10^6$ cells were electroporated using a Neon Transfection kit (ThermoFisher Scientific) and re-suspended in ribonucleoprotein (RNP) complex. The RNP was made by incubating a gRNA targeting the first exon of *WAS* and High Fidelity Cas9 protein (Integrative DNA Technologies) at a molar ratio of 1:2 at 37°C for 15 min. The conditions for electroporation were 1600V, 10

ms, and 3 pulses. Following electroporation, cells were seeded at concentration of $1 \times 10^6$ cells per ml and incubated at 37℃ for 15 min after which an adeno-associated virus 6 (AAV6) donor vector containing a *WAS* cDNA flanked by *WAS* homology arms was added at 50,000 MOI (vector genomes/cell) for 12 hr. After that, cells were washed twice with PBS to remove viral particles and cultured in monocyte differentiation media.

## Lentiviral transduction of WAS HSPCs

Two days after thawing, WAS HSPCs were transduced with a clinical grade WW1.6 lentiviral vector produced by Genethon (*Hacein-Bey Abina et al., 2015*) at an MOI of 100 for 12 hr. After that, cells were washed twice with PBS to remove viral particles and cultured in monocyte differentiation media.

## Differentiation of CD34⁺ HSPCs into macrophages in vitro

Patient-derived WAS CD34⁺ HSPCs were seeded at a density of 40,000 cells per $cm^2$ in IMDM (ThermoFisher Scientific) supplemented with SCF (20 ng/ml), FLT3-ligand (30 ng/ml), IL-3 (30 ng/ml), M-CSF (30 ng/ml, ThermoFisher Scientific), FCS (20%, ThermoFisher Scientific) at 37℃ / 5% $CO_2$. After 7 days in culture, the cells were isolated for CD14+ monocytes using CD14+ Microbead kit (Miltenyi Biotec) according to the manufacturer's protocol. Purified CD14+ monocytes were cultured in RPMI (supplemented as for THP-1 and MDMs) containing M-CSF 20 ng/ml for a further 5–7 days.

## Determination of vector copy number

Digital Droplet PCR (ddPCR) was performed to measure vector copy number (VCN). Briefly, genomic DNA was extracted 2 weeks post-transduction using DNeasy Blood and Tissue extraction kit (Qiagen), according to manufacturer's protocol. In a total volume of 22 µl ddPCR, 20 ng of genomic DNA was combined with 10 µM each of target primer and FAM probe mix, 10 µM each of reference primer and HEX probe mix, $1 \times$ ddPCR Supermix probe without dUTP (Bio-Rad) and nuclease free water. Individual droplets were generated using QX100 Droplet Generator (Bio-Rad) and subsequently amplified in a Bio-Rad PCR thermocycler. The optimised amplification steps were: Step 1 – 95℃ for 10 min; Step 2 (49 cycles) - 94℃ for 1 min, 60℃ for 30 s, 72℃ for 2 min; Step 3–98℃ for 10 min. The Droplet Reader and QuantaSoft Software (both from Bio-Rad) were used to record and analyse the positive and negative fluorescence droplets according to the manufacturer's guidelines (Bio-Rad). VCN was calculated as the ratio of FAM to HEX signal after normalization against the reference signal.

## Detection of WASp expression

Detection of intracellular WASp in CD14⁺ monocytes was carried out by fixing cells with 4% paraformaldehyde (PFA) followed by incubation with primary mouse anti-human WASP antibody (Clone 5A5, BD Bioscience) in 0.1% Triton X-100. Cells were stained with secondary goat anti-mouse IgG Alexa Fluor 647 antibody (clone Poly4053, BioLegend) to determine percentage of WASp-positive cells by FACS using a BD LSRII instrument (BD Bioscience).

## Reagents

For non-selective autophagy induction and inhibition of autophagosome-lysosome fusion, rapamycin (Calbiochem) and bafilomycin A1 (Sigma) were used at concentrations of 50 nM and 160 nM respectively. CK666 (Abcam) was used to inhibit ARP2/3 in primary MDMs at 20 µM or 100 µM as indicated. CCCP (Merck) to induce mitophagy was used at a concentration of 10 µM for 2 hr.

## Immunoblotting

Cells were seeded in 12 well plates at a density of $0.5 \times 10^6$ cells/ well and lysed in RIPA buffer (Sigma) containing protease (Roche) and phosphatase inhibitor (Sigma) cocktails after completion of experiments. Protein lysates were reduced with DTT (Thermo Fisher Scientific), heat blocked and separated by SDS-PAGE using 4–12% pre-cast BisTris gels (Thermo Fisher Scientific) with MES running buffer (Thermo Fisher Scientific), using NuPAGE (Invitrogen) system. Resolved proteins were transferred to methanol-activated 0.2 µm PVDF membrane (BioRad), using BioRad Mini Transblot Cell wet transfer system (100V for 1 hr) and transfer buffer (Thermo Fisher Scientific) with 20%

methanol. After blocking in 5% skimmed milk for 1 hr at room temperature, membranes were incubated with primary antibodies [LAMP-1 (1:1000, BD Biosciences), LC3B (1:1000, Sigma) and WASp (1:500, BD Biosciences) as needed, with GAPDH (1:1000, Santa Cruz) as loading control] for either 1 hr at room temperature or overnight at 4°C. Anti-mouse and anti-rabbit HRP-conjugated secondary antibodies were used at concentrations of 1:2000 (GE healthcare) for 1 hr at room temperature and membranes developed after incubating for 2 min with SuperSignal West Pico Chemiluminescent Substrate (Thermo Fisher Scientific). Where needed, membranes were stripped with ReBlot Plus Strong Antibody Stripping Solution (EMD Millipore), according to manufacturer's instructions. Densitometries were calculated using ImageJ software.

## Immunofluorescence and confocal microscopy

Cells were seeded onto borosilicate 1.5, 13 mm cover slips (VWR) in 24-well plates at a density of $0.2 \times 10^6$ cells/well. For podosome analysis, cover slips were pre-treated with fibronectin 10 µg/ml (R and D Systems) overnight before seeding cells. At the end of experiments, cells were washed twice in ice-cold PBS and fixed using 4% chilled paraformaldehyde (Thermo Fisher Scientific) for 10 min on ice, followed by 8% at room temperature for 20 min. Cells were permeabilised with 0.1% triton X-100 (Sigma) for 5 min at room temperature and non-specific binding reduced by blocking with 5% BSA in PBS with 5% goat serum (Cell Signalling Technology) for 1 hr at room temperature. Primary antibodies were diluted in 5% BSA/PBS [LC3 (1:200, MBL), LAMP-1 (1:50, CST), vinculin (1:200, Sigma)] and cells incubated for 1 hr at room temperature, followed by incubation with secondary antibodies-conjugated to fluorochromes (all Molecular Probes) [anti-mouse-488 (1:500), anti-rabbit 547 (1:500), anti-rabbit-647 (1:500)] for 1 hr at room temperature. F-actin was labelled with phalloidin-633 (1:200, Molecular Probes). Cover slips were mounted onto Superfrost microscope slides (Thermo Fisher Scientific) using Prolong Diamond anti-fade mounting solution with DAPI (Molecular Probes). Slides were blinded prior to imaging and analysis as indicated in figure legends. Fluorescence microscopy images were acquired using an inverted Zeiss LSM 710 confocal microscope at 20x or 63x as indicated. Equal numbers of images from sequential fields of view of each blinded slide were saved and analysed using ImageJ before sample identities were released. Super-resolution images were obtained by deconvolving confocal images using Huygens software.

## Transmission electron microscopy

Cells were seeded at a density of $1 \times 10^6$ cells/well into Corning Falcon Easy Grip 35-mm tissue culture dishes (Thermo Fisher Scientific). After experiments, cells were washed in ice cold PBS, fixed in 0.5% glutaraldehyde/200 mM sodium cacodylate pH 7.2 and prepared/images acquired as previously described (*Lee et al., 2017*).

## Identification of potential LC3-interacting domains

Potential LC3-interacting regions (LIRs) were identified using the iLIR software resource as described in *Kalvari et al., 2014*. The software analyses protein sequences for simple four amino acid combinations starting with tryptophan (W) and ending with leucine (L) (WxxL), which they describe as being the shortest sequence required for possible interaction with an Atg8-family protein. Additionally, the team identified extended potential LIR motifs (xLIR) of six amino acid sequences, which they demonstrate confer greater sensitivity and specificity of identifying true LIR domains. Position-specific scoring matrix (PSSM) scores are quoted, which compare the similarity of potential new LIR domains with those already validated. The score ranges from −20 to +20 and has been validated for known sequences. A cut off of 9 is used to suggest possible LIR identification, where sensitivity was found to be high, but specificity very low. With increasing PSSM score, the sensitivity decreases, whereas specificity increases. Meaningful PSSM scores are identified in the range 13–17, with balanced accuracy most optimal at 15 or 16.

## Extracellular metabolic flux analysis

A Seahorse XF$^e$96 metabolic extracellular flux analyser was used to determine the extracellular acidification rate (ECAR) in mpH/min and the oxygen consumption rate (OCR) in pmol/min. In brief, frozen PBMCs were thawed and processed as described above and seeded (according to availability, 0.5 or $1 \times 10^5$/well in respective experiments) in serum-free unbuffered RPMI 1640 medium (Sigma

Aldrich #R6504) onto Cell-Tak (#354240, Coring, NY, USA) coated cell plates. Mito Stress test was performed by sequential addition of oligomycin (1 µM; Sigma Aldrich 75351), carbonyl cyanide-4-(tri-fluoromethoxy)phenylhydrazone (FCCP; 2 µM; Sigma Adlrich C2920) and rotenone (1 µM; Sigma Aldrich R8875) at the indicated time points. Metabolic parameters were calculated as described previously (*Gubser et al., 2013*).

## Statistical analysis

All statistical analyses were undertaken using GraphPad Prism. Data presented are mean ± standard error of the mean (SEM) with two-tailed unpaired t-test unless otherwise stated.

## Acknowledgements

This work was supported by funding from The Wellcome Trust (090233/Z/09/Z AJT and 201250/Z/16/Z ER) and National Institute for Health Research Biomedical Research Centre at Great Ormond Street Hospital for Children NHS Foundation Trust and University College London.

## Additional information

### Funding

| Funder | Grant reference number | Author |
|---|---|---|
| Wellcome Trust | 090233/Z/09/Z | Adrian James Thrasher |
| Wellcome Trust | 201250/Z/16/Z | Elizabeth Rivers |
| National Institute for Health Research | Great Ormond Street Hospital for Children NHS Foundation Trust | Elizabeth Rivers<br>Rajeev Rai<br>Alessia Cavazza<br>Mona Bajaj-Elliott<br>Adrian J Thrasher |

The funders had no role in study design, data collection and interpretation, or the decision to submit the work for publication.

### Author contributions

Elizabeth Rivers, Conceptualization, Funding acquisition, Data curation, Formal analysis, Supervision, Validation, Investigation, Writing - original draft, Writing - review and editing; Rajeev Rai, Investigation; Jonas Lötscher, Michael Hollinshead, Investigation, Writing - review and editing; Gasper Markelj, James Thaventhiran, Austen Worth, Resources, Writing - review and editing; Alessia Cavazza, Supervision, Methodology, Writing - review and editing; Christoph Hess, Supervision, Writing - review and editing; Mona Bajaj-Elliott, Conceptualization, Supervision, Writing - review and editing; Adrian J Thrasher, Conceptualization, Funding acquisition, Writing - review and editing

### Author ORCIDs

Elizabeth Rivers ⓘ https://orcid.org/0000-0001-5814-8014
Austen Worth ⓘ https://orcid.org/0000-0001-6803-7385
Adrian J Thrasher ⓘ https://orcid.org/0000-0002-6097-6115

### Ethics

Human subjects: For usage of human CD34+ HSPC from healthy and WAS donors, informed written consent was obtained in accordance with the Declaration of Helsinki and ethical approval from the Great Ormond Street Hospital for Children NHS Foundation Trust and the Institute of Child Health Research Ethics (08/H0713/87).

### Decision letter and Author response

Decision letter https://doi.org/10.7554/eLife.55547.sa1
Author response https://doi.org/10.7554/eLife.55547.sa2

## Additional files

### Supplementary files
• Transparent reporting form

### Data availability
All data associated with this study are present in this manuscript and Supporting Files.

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
