## [Decision Letter]

**Acceptance summary:**

This study shows that WASp regulates autophagy and mitophagy in immune cells, including primary monocyte/HSPC-derived macrophages from WAS patients, of relevance to Wiskott Aldrich Syndrome.

**Decision letter after peer review:**

Thank you for submitting your article "Wiskott Aldrich syndrome protein regulates non-selective autophagy and mitochondrial homeostasis in human myeloid cells" for consideration by *eLife*. Your article has been reviewed by three peer reviewers, one of whom is a member of our Board of Reviewing Editors, and the evaluation has been overseen by Tadatsugu Taniguchi as the Senior Editor. The reviewers have opted to remain anonymous.

The reviewers have discussed the reviews with one another and the Reviewing Editor has drafted this decision to help you prepare a revised submission.

Summary:

In this manuscript, the authors investigate the role of WASp in autophagy and mitophagy in immune cells, including primary monocyte/HSPC-derived macrophages from WAS patients, of clear relevance to WAS. Using a model of rapamycin-induced autophagy, the authors show that WASp plays a role in autophagy induction, as indicated by LC3 punctae formation and LC3 lipidation. In SDMs and MDMs from WAS patients, such defects in autophagy induction could be corrected by reconstitution with wild-type WASp, or CRISPR/Cas9-mediated gene editing. The role of WASp in autophagy was mediated by the WASp effector ARP2/3, since chemical or genetic block of ARP2/3 leads to defects in autophagy, suggesting the involvement of actin cytoskeleton in the processes. WASp and ARP2/3 also regulate mitophagy, mediating the encapsulation of mitochondria in actin cages during mitophagy, and maintenance of mitochondrial networks. Finally, mitochondrial function (oxidative metabolism) was impaired in MDMs from WAS patients, as indicated by Seahorse.

While the authors' findings suggest how WASp deficiency can perturb autophagy and mitophagy to impair immune cell functions in WAS, there are some concerns that should be addressed before publication. Most notably, key controls are missing from many experiments that make data interpretation difficult. Also, it would also be nice if the authors could provide a bit more mechanistic insight given that some of what they have shown regarding WASp has already been described in other cell types.

Essential revisions:

1) In many parts of the paper, the authors are not consistent in presenting data with controls. For example, in Figure 1, while there are untreated controls for both WT THP-1 and WAS KO THP-1 cells in Figure 1A, there is no control for HC cells in Figure 1C and no control at all in Figure 1E and G. In many experiments, different stimulation conditions appear to have been used on cells from the healthy donor and the ARPC1B patient, which makes the results not strictly comparable. For example, in Figure 3E, conditions for healthy donor and ARPC1b patient are not the same.

2) Previous studies by the same lab and other labs have already shown that WASP and other WASP family proteins including JMY and WHAMM are involved in autophagosome formation. In all cases including the data from the current study, activation of ARP2/3 is required for mediating their effect. It is however unclear whether they use similar mechanisms to initiate the autophagy process. Is it possible that WASP, like JMY, possess LC3-interacting domain and so can be recruited to LC3-containing substrates/cargos for autophagy? Can the authors stain WASP and examine colocalization with LC3? These experiments would also provide more mechanistic insight into some of the authors' findings.

3) The authors show that MDMs from WAS patients have impaired oxidative metabolism as assessed by Seahorse. Does reconstitution with wild-type WASp and/or CRISPR/Cas9-mediated gene editing rescue this defect?

4) The authors described three main categories of mitochondrial morphology: networked, elongated and fragmented (Figure 5A). Because in many cases elongated mitochondria are considered to be networked, can the authors provide justification from their own experiments or from the literature indicating a functional distinction between networked and elongated mitochondria? The authors should also describe how they scored fragmented, networked, or elongated mitochondria.

---

## [Author Response]

Essential revisions:1) In many parts of the paper, the authors are not consistent in presenting data with controls. For example, in Figure 1, while there are untreated controls for both WT THP-1 and WAS KO THP-1 cells in Figure 1A, there is no control for HC cells in Figure 1C and no control at all in Figure 1E and G. In many experiments, different stimulation conditions appear to have been used on cells from the healthy donor and the ARPC1B patient, which makes the results not strictly comparable. For example, in Figure 3E, conditions for healthy donor and ARPC1b patient are not the same.

Unfortunately for Figure 1C the number of CD14+ cells isolated from 5mls of healthy donor PBMCs was much lower compared with the number of cells isolated from the WAS patient and enough cells were only available for treated conditions in the healthy donor. Due to the nature of WAS as a rare disease, access to large number of patient primary cells needed for western blotting is very limited and repeating these experiments would therefore be extremely challenging. For simplicity, this figure has now been revised to display comparable data between the healthy donor and WAS patient (revised Figure 1C). Furthermore, the reduced LC3II protein expression demonstrated in Figure 1C was presented only to support the data from cell lines and is further demonstrated in the LC3 punctae formation visualised by confocal microscopy, as shown in revised Figure 1G.

For Figure 1E and G, controls were not presented purely to reduce the figure size. Representative images have now been added and can be viewed in revised Figure 1E and G.

The stimulation conditions used for confocal imaging of primary cells were always the same. We thank the reviewer for picking up a typo in the display for Figure 3E, which should have noted the same conditions for healthy donor and ARPC1B patient, reflecting the true experimental conditions. This figure has now been revised accordingly (revised Figure 3E).

2) Previous studies by the same lab and other labs have already shown that WASP and other WASP family proteins including JMY and WHAMM are involved in autophagosome formation. In all cases including the data from the current study, activation of ARP2/3 is required for mediating their effect. It is however unclear whether they use similar mechanisms to initiate the autophagy process. Is it possible that WASP, like JMY, possess LC3-interacting domain and so can be recruited to LC3-containing substrates/cargos for autophagy? Can the authors stain WASP and examine colocalization with LC3? These experiments would also provide more mechanistic insight into some of the authors' findings.

We believe that the mechanism of WASp action in autophagy and mitophagy is through its interaction with ARP2/3. Rapamycin-induced autophagy stimulation of primary MDMs did not find consistent localisation of WASp with LC3 (Author response image 1). Intracellular staining of WASp with other proteins is notoriously unreliable, even for established WASp functions such as podosome formation. However, analysis of LC3 punctae in eGFP-WASp THP-1 cells also did not demonstrate colocalization between the two (Author response image 1). We know from Figures 1, 3 and 4 that autophagy is impaired in the absence of either WASp or ARP2/3, suggesting that the interaction of the two is crucial. Further imaging looking for ARP2/3 in addition to WASp was not pursued since the role of ARP2/3 and WASp in autophagosome formation may be transient and therefore better captured in a dynamic imaging model. Additionally, the finding of ARP2/3 and/or WASp in complex with LC3 in rapamycin-induced autophagy would not necessarily infer insight into functionality of these interactions. Having said this, our previous work did identify WASp in complex with actin and septins in xenophagy (Lee et al., 2017). It is possible that WASp and/or ARP2/3 might be more easily identified with LC3 in our mitophagy model, but again, further information on the functionality of these interactions would not be gained through confocal imaging.

**Author response image 1. sa2fig1:** a) Healthy donor MDMs cultured with rapamycin and bafilomycin for 6 hours. Fixed and stained for LC3 (red), WASp (green) and DAPI (blue). Analysed by confocal microscopy at 63x. Panel A displays an example of LC3 punctae without WASp localisation. Panel B displays an example where LC3 appears to be localising with WASp. b) eGFP-WASp THP-1 cells cultured with rapamycin and bafilomycin for 6 hours. Fixed and stained for LC3 (red), nuclei (DAPI). Analysed by confocal microscopy at 63x.

However, to explore further our hypothesis that the role of WASp in autophagy and mitophagy is through its interaction with ARP2/3, we used the iLIR web resource platform developed by Kalvari et al. (Kalvari et al., 2014), to analyse the likelihood of an LC3-interacting region (LIR) in WASp. The software analyses protein sequences for simple four amino acid combinations starting with tryptophan (W) and ending with leucine (L) (WxxL), which they describe as being the shortest sequence required for possible interaction with an Atg8-family protein. Additionally, the team identified extended potential LIR motifs (xLIR) of 6 amino acid sequences, which they demonstrate confer greater sensitivity and specificity of identifying true LIR domains. Position-specific scoring matrix (PSSM) scores are quoted, which compare the similarity of potential new LIR domains with those already validated. The score ranges from -20 to +20 and has been validated for known sequences. A cut off of 9 is used to suggest possible LIR identification, where sensitivity was found to be high, but specificity very low. With increasing PSSM score, the sensitivity decreases whilst specificity increases. Meaningful PSSM scores are identified in the range 13-17, with balanced accuracy most optimal at 15 or 16.

Analysis of WASp’s amino acid sequence using this web resource did not find any xLIR domains (Figure 3—source data 1). Several WxxL motifs were identified, but PSSM scores were low (max. 10), making a functional interaction between WASp and LC3 unlikely. The same process was carried out for all seven ARP2/3 complex subunits. In contrast to WASp analysis, three out of the seven ARP2/3 subunits (ARP2, ARP3 and ARPC2) were found to contain xLIR motifs, with high PSSM scores, highly predictive of functional LIR domains. Additionally, ARPC1B was identified to have a WxxL motif with high PSSM score that could also be predicted to contain a functional LIR motif.

Whilst the above analysis would support our hypothesis that WASp is acting through ARP2/3 in autophagy, it is also worth mentioning that LIR motif-independent modes of interaction with Atg-8 family proteins have also been reported (Behrends et al., 2010).

3) The authors show that MDMs from WAS patients have impaired oxidative metabolism as assessed by Seahorse. Does reconstitution with wild-type WASp and/or CRISPR/Cas9-mediated gene editing rescue this defect?

We are very interested to study this, but have been limited by reduced face-to-face contact with patients, access to primary cells and lab availability in the current environment. However, recovery of mitochondrial morphology, as seen in revised Figure 5B, would allow us to speculate that this may be the case.

4) The authors described three main categories of mitochondrial morphology: networked, elongated and fragmented (Figure 5A). Because in many cases elongated mitochondria are considered to be networked, can the authors provide justification from their own experiments or from the literature indicating a functional distinction between networked and elongated mitochondria? The authors should also describe how they scored fragmented, networked, or elongated mitochondria.

Classification of mitochondrial morphology was based on visual appearances. It is unclear whether there is a functional distinction between networked and elongated mitochondria, but a visual difference was observed, where mitochondria appear more joined together in some cells. We wondered if that could reflect mitochondria in a different state and where found in increased proportion (e.g. ARPC1B-deficient cells) could indicate a potential difference in mitochondrial fission more than fusion. As the reviewer points out, elongated mitochondria are considered to be a form of networked and so figures have been revised to combine elongated and networked morphologies (revised Figure 5A and C).